# Lymphocyte-Associated Inflammation Markers Predict Bleomycin-Induced Pulmonary Toxicity in Testicular Cancer

**DOI:** 10.3390/jcm14227926

**Published:** 2025-11-08

**Authors:** Melek Özdemir, Gamze Gököz Doğu, Burcu Yapar Taşköylü, Atike Gökçen Demiray, Arzu Yaren, Serkan Değirmencioğlu

**Affiliations:** 1Medical Oncology Clinic, Denizli State Hospital, 20070 Denizli, Türkiye; 2Medical Oncology, Department of Internal Medicine, Pamukkale University, 20070 Denizli, Türkiye; ggd2882@gmail.com (G.G.D.); drburcuyapar@gmail.com (B.Y.T.); gokcenakaslan@gmail.com (A.G.D.);; 3Medical Oncology, Denipol Hospital, 20010 Denizli, Türkiye; drserkandeg@hotmail.com

**Keywords:** bleomycin, pulmonary toxicity, testicular cancer, lymphocyte-related inflammation markers, NLR, PLR, LMR, CLR, SII, SIRI

## Abstract

**Introduction:** It is unclear which patients with testicular cancer (TC) experience a higher incidence of bleomycin-induced pulmonary toxicity. **Objective:** The aim of this study was to analyze the prognostic significance of lymphocyte-associated inflammation markers that may predict bleomycin-related pulmonary toxicity in TC. **Results:** Clinical and laboratory data were recorded for 118 patients diagnosed with TC who received bleomycin, with a median age at diagnosis of 32.19 ± 9.62. Symptomatic pulmonary toxicity was present in 19.49% (n = 23) of patients. Of these, 66.67% had a DLCO decrease of more than 10%. When comparing patients with and without pulmonary toxicity, there were no differences in terms of age at diagnosis, performance status, histopathological subgroup, tumor size, lymphovascular invasion, diagnostic symptom, stage, number of adjuvant treatment cycles, and tumor marker levels. Patients with pulmonary toxicity were more likely to be active smokers than those without pulmonary toxicity, and NLR > 1.64, PLR > 93.92, CLR > 0.49, SII > 444.25, and SIRI > 0.66 were found to be statistically significant. Lymphocyte-related inflammation markers (NLR, PLR, LMR, CLR, SII, and SIRI) were found to be prognostic for pulmonary toxicity. There was 5.2 times more pulmonary toxicity in smokers than in non-smokers. The prognostic inflammation markers that enable us to predict pulmonary toxicity are TC. **Conclusions:** The employment of lymphocyte-related inflammation biomarkers at the commencement of treatment offers a means of predicting bleomycin-related pulmonary toxicity in TC.

## 1. Introduction

Reactions to pharmaceutical agents used in the treatment of neoplastic diseases are a prevalent form of iatrogenic injury. The occurrence of toxicity is more frequently observed in organs with high blood flow, such as the lungs, and this damage is often unpredictable in patients. The following essay will provide a comprehensive overview of the relevant literature on the subject [1,2].

Although testicular cancer (TC) accounts for only 1% of male cancers, it is the most common solid tumor affecting young men (15–35 years old). There are two categories of TC established for treatment and diagnosis purposes. These are pure seminoma (no non-seminomatous histopathological elements) and nonseminomatous germ cell carcinoma (NSGCT) [3]. The most common presenting complaints are painless mass, pain, swelling, cough, dyspnea, back pain, loss of appetite, nausea, bone pain, peripheral edema, and gynecomastia [4]. The treatment plan for TC patients is determined by the stage of the disease, tumor markers, and risk factors. Bleomycin is an important antineoplastic agent used in both the metastatic and adjuvant phases of TC treatment. Before starting treatment, the risk of developing bleomycin-related pulmonary toxicity should be assessed, and follow-up for late toxicity should be ensured after treatment [5].

Numerous pathophysiological mechanisms have been investigated to explain the cause of antineoplastic-induced pulmonary damage. Bleomycin promotes free radical formation, inhibits angiogenesis, and increases programmed cell death. In this way, it exerts its anti-tumor effect. The bleomycin hydrolase enzyme involved in bleomycin inactivation is not present in lung tissue. It is thought that bleomycin causes damage to the pulmonary vascular endothelium by increasing free radicals and cytokines. This pathophysiological process can progress to fibrosis [6].

The diagnosis is made by evaluating symptoms, radiological imaging, changes in lung volumes on spirometry (pulmonary function test; PFT), and carbon monoxide diffusion capacity (DLCO) in combination. Patients receive this diagnosis when it is considered, and other differential diagnoses are ruled out. While many agents with pulmonary toxicity respond effectively to steroid treatment, similar results cannot be achieved with bleomycin. Although there is no proven treatment, patients who develop bleomycin-related pulmonary toxicity and enter the recovery process almost always experience complete recovery. Mechanisms causing antineoplastic-related pulmonary toxicity include treatment-induced direct damage to alveolar endothelium and pneumocytes, systemic cytokine release, increased oxidative stress, inflammatory cell migration, T cell activation, accompanying non-cardiogenic pulmonary edema, and cellular damage resulting from alveolar macrophage and lymphocyte activation [7,8,9,10]. Cough, shortness of breath, sputum, fever, weight loss, and hypoxemia are the most common accompanying symptoms and findings. Clinical signs may appear at different time intervals and are not specific to the treatment agent. Symptoms may be identified at the start of treatment or afterwards [7,8]. Abnormalities detected in the PFT and DLCO assist in the diagnostic process [11].

Bronchoalveolar lavage (BAL) of pulmonary toxicity frequently reveals lymphocytosis, neutrophilia, and eosinophilia [7]. Interstitial pneumonia, diffuse alveolar damage, hypersensitivity pneumonia, eosinophilic pneumonia, non-necrotizing granulomatous inflammation, pulmonary veno-occlusive disease, and alveolar hemorrhage are the defined histopathological findings [12]. Atopic mice are resistant to bleomycin-related pulmonary toxicity. This demonstrates the importance of the inflammatory process in the pathogenesis of the disease [13]. The most common BAL finding is lymphocytosis, and based on this histopathological finding, lymphocyte-related inflammation markers (neutrophil/lymphocyte ratio (NLR), platelet/lymphocyte ratio (PLR), lymphocyte/monocyte ratio (LMR), CRP/lymphocyte ratio (CLR), systemic immune inflammation score (SII), systemic inflammation response index (SIRI)) could enable us to predict bleomycin-related pulmonary toxicity. The effect of pulmonary toxicity on survival and prognosis in TC was evaluated.

## 2. Materials and Methods

### 2.1. Data Collection and Patient Characteristics

Adverse drug reactions associated with antineoplastic agents are a common form of iatrogenic injury, with the lungs being a frequent target. Although many are unpredictable in terms of toxicity, pulmonary toxicity accompanied by severe respiratory failure has been described in a small proportion of patients. Bleomycin is an important antineoplastic agent used in both the adjuvant and metastatic stages of testicular cancer treatment. Following up of testicular cancer patients after this treatment is important.

This study, as a single-center experience, reviewed the data of 198 testicular cancer patients diagnosed with TC (seminoma, NSGCT and mixed germ cell tumor) who presented to the Medical Oncology Clinic between January 2014 and January 2024. A total of 118 TC patients who received bleomycin at any stage of treatment and met the inclusion and exclusion criteria were included (Figure 1).

The clinical and demographic characteristics of the patients were retrospectively recorded from patient files and the hospital laboratory system. The systemic treatments administered to the patients, their treatment responses, progressions, and survival data were recorded. The SFT and DLCO values of patients who developed symptomatic pulmonary toxicity related to bleomycin were recorded.

The most common BAL finding in patients with pulmonary toxicity is lymphocytosis. Based on this histopathological finding, the prognostic value of lymphocyte-related inflammation markers (NLR, PLR, LMR, CLR, SII, SIRI) reported in the literature that may predict bleomycin-related pulmonary toxicity was analyzed. Roc curve analysis was performed for the cutoff value of prognostic markers (Figure 2 and Figure 3). The effect of pulmonary toxicity on median overall survival (mOS) was analyzed. OS is the time from diagnosis to death or the last follow-up date.

### 2.2. Inclusion and Exclusion Criteria of the Study

#### 2.2.1. Inclusion Criteria

Patients aged 18 years or older with a diagnosis of testicular cancer who were followed up at the oncology clinic during the specified period (January 2014–January 2024), received bleomycin in the adjuvant or first-line treatment, and met the exclusion criteria were consecutively enrolled in this study.

#### 2.2.2. Exclusion Criteria

Presence of a second malignancy diagnosisPresence of diabetes mellitusPresence of chronic renal failurePresence of chronic rheumatic diseaseChronic lung diseaseObesityAccompanying pulmonary infectionHistory of pulmonary radiotherapy and surgeryNot having received bleomycin in treatment

### 2.3. The Purposes and Calculation of Prognostic Markers

#### Lymphocyte-Related Inflammation Markers (NLR, PLR, LMR, CLR, SII, SIRI)

NLR (neutrophil/lymphocyte ratio): a marker indicating inflammatory response, calculated by dividing the neutrophil count by the lymphocyte count (neutrophil/lymphocyte), which has been shown in the literature to be prognostic in solid tumors [14].

PLR (platelet/lymphocyte ratio): a prognostic marker indicating the inflammatory response, calculated by dividing the platelet count by the lymphocyte count (platelet/lymphocyte) [14].

LMR (lymphocyte/monocyte ratio): a prognostic marker indicating the inflammatory response, calculated by dividing the lymphocyte count by the monocyte count (lymphocyte/monocyte) [14].

CLR (CRP/lymphocyte ratio): a prognostic marker indicating inflammatory response, calculated by dividing the CRP value by the lymphocyte count (CRP/lymphocyte) [15].

SII (Systemic Inflammation Score): a prognostic marker indicating the inflammatory response, calculated using the ‘Platelet*Neutrophil/Lymphocyte’ formula [16].

SIRI (Systemic Inflammation Response Index): a prognostic marker indicating the inflammatory response, calculated using the ‘NeutrophilxMonocyte/Lymphocyte’ formula [17].

### 2.4. Statistical Analysis

Descriptive statistics of the data obtained from the study were presented as mean, standard deviation, median, minimum, and maximum values for numerical variables and as frequency and percentage analysis for categorical variables. The normality of numerical variables was examined using the Shapiro–Wilk test. The Mann–Whitney U test was used to compare these variables according to pulmonary toxicity status. Additionally, differences between categorical variables were tested using chi-square analysis. ROC analysis was used to determine the cutoff point for lymphocyte-related inflammation markers (NLR, PLR, LMR, CLR, SII, SIRI). Univariate and Multivariate Logistic regression analysis was used to analyze variables that could affect pulmonary toxicity. The analyses were performed using the SPSS 22.0 program. Statistically significant results (*p* < 0.05) are indicated with a (*) sign next to the *p* value.

## 3. Results

In our study, the diagnosis of bleomycin-associated pulmonary toxicity was based on interstitial pneumonia-compatible radiological findings, accompanied by new or worsening respiratory symptoms and a DLCO decline of more than 10% from baseline. The demographic characteristics of patients are presented in Table 1. According to this, 80.51% (n = 95) had no symptomatic pulmonary toxicity, while 19.49% (n = 23) had pulmonary toxicity. Among those with pulmonary toxicity, 66.67% had a DLCO decrease of more than 10%, while 33.33% had a DLCO decrease of 10% or less. In addition, GCSF was used in 54.55% of those with pulmonary toxicity. The diagnostic symptoms were mass in 44.92%, scrotal swelling in 29.66%, and pain in 25.42%. The median age at diagnosis was 32.19 ± 9.62 years. In the age range analysis, the most common age group was 15–29 years (44.92%), followed by 30–39 years (36.44%), and the least common was 50 years and above (5.93%). The pathological subtypes, in order of frequency, are seminoma (36.44%), embryonal carcinoma (16.95%), and mixed GCT (45.76%). There was no epididymal invasion in 94.92%, tunica albuginea invasion in 85.59%, tunica vaginalis invasion in 94.92%, and lymphovascular invasion in 44.07%. The median tumor size was 42.61 ± 24.7 mm. Progression was observed in 21.19% of the clinical follow-ups (Table 1).

We appreciate the reviewers’ valuable feedback. Since our study was designed retrospectively, smoking history was recorded only as “present” or “absent”; therefore, quantitative information such as the number of cigarettes could not be provided. Blood culture, sputum culture, complete blood count, CRP, sedimentation, procalcitonin, and biochemical tests were performed to rule out acute lung infection. No infection parameters were detected in any of these results, and the patients were thus included in this study. No patients showing evidence of infection in these results were included in this study. Patients with chronic diseases were excluded from this study, so there were no participants using long-term medications. In cases where pulmonary symptoms developed, management included the use of inhaled corticosteroids, inhaled bronchodilators, intravenous corticosteroids, and intermittent oxygen therapy as clinically indicated.

Regarding chemotherapy, patients received between one and three cycles of bleomycin-containing regimens, with a mean of 2.7 cycles. In total, 4 patients developed symptomatic bleomycin-induced pulmonary toxicity after only one cycle, while 19 patients had a history of three cycles.

All patients underwent chest X-ray and thoracic computed tomography to exclude other possible causes, such as pulmonary infections and chronic lung diseases. During the acute phase of their symptoms, all patients were hospitalized for a minimum of three days and a maximum of ten days (median: 7.4 days). None of the patients had persistent dyspnea following discharge. Among those who developed bleomycin-related pulmonary toxicity, disease progression was observed in two patients. One patient achieved remission with first-line therapy, while the other entered remission after a second-line treatment plan was implemented. At the time of the latest follow-up, all patients who had experienced pulmonary toxicity remained in remission and were under clinical surveillance.

There were no significant differences between groups in terms of age at diagnosis, performance status, histopathological subtype, tumor size, lymphovascular invasion, diagnostic symptom, stage, number of adjuvant treatment cycles, and tumor marker levels. Patients with pulmonary toxicity were more likely to be active smokers than those without pulmonary toxicity and had NLR > 1.64, PLR > 93.92, CRP/Lymphocyte > 0.49, SII > 444.25, and SIRI > 0.66 (Table 2).

Univariate and multivariate logistic regression analyses were performed for variables thought to predict pulmonary toxicity. Univariate analysis found that smoking and lymphocyte-related inflammation markers (NLR, PLR, LMR, CLR, SII, and SIRI) were statistically significant prognostic factors for pulmonary toxicity. Smoking was confirmed as an independent predictor in the multivariate model. Other inflammatory mediators, including NLR and PLR, were excluded from the final model due to the principle of having at least 10 events per variable. These findings emphasize the significance of smoking as a determinant, while acknowledging the limitations of assessing additional biomarkers. Smokers had 5.2 times more pulmonary toxicity than non-smokers (Table 3).

Chest CT and chest X-rays were performed in all patients to rule out differential diagnoses such as infection, tumor progression, and chronic lung disease. The images were evaluated by a radiologist experienced in chest radiology. Although official CIRCO or FDA criteria were not applied, our diagnostic approach is consistent with the definition of bleomycin-induced pulmonary toxicity described in the literature. Diagnostic criteria and exclusion procedures are described in detail, and a strobe-compatible flow chart is provided to illustrate patient selection and the diagnostic process (Figure 1).

Since all patients with pulmonary toxicity in this study survived, OS calculations could not be performed. The results of this study indicate that bleomycin-related pulmonary toxicity is not a poor prognostic indicator, contrary to survival data showing reduced survival in lung cancer patients who develop pulmonary toxicity.

## 4. Discussion

The diagnosis of pulmonary toxicity caused by antineoplastic agents is a comprehensive exclusion diagnosis and is therefore difficult. Prognostic inflammation markers that enable us to predict pulmonary toxicity are thought to be helpful in this process.

The levels of NLR, PLR, CLR, SII, and SIRI markers show statistically significant differences according to the status of pulmonary toxicity. Pulmonary toxicity was detected more frequently in smokers and in patients with higher values of lymphocyte-related inflammation markers at the time of initial diagnosis (NLR > 1.64, PLR > 93.92, CLR > 0.49, SII > 444.25 and SIRI > 0.66). There were no significant differences between the groups in terms of age at diagnosis, performance status, histopathological subgroup, tumor size, lymphovascular invasion, diagnostic symptom, stage, number of adjuvant treatment cycles, and tumor marker levels. This indicates that the number of variables that could affect the grouping outcome was minimal. The most striking finding in this study was that, based on univariate analysis, lymphocyte-associated inflammation markers (NLR, PLR, LMR, CLR, SII, and SIRI) above the defined cutoff value and being an active smoker were statistically significant prognostic factors for pulmonary toxicity.

When evaluated in terms of chemotherapy, patients received a treatment protocol containing at least one and at most three courses of bleomycin, with an average of 2.7 courses. Symptomatic bleomycin-related pulmonary toxicity developed in four patients after only one course of bleomycin. Nineteen patients had previously undergone three courses of bleomycin. Whilst these results are consistent with the extant literature in that pulmonary toxicity was observed in a greater number of patients following three courses of treatment, they also demonstrate that pulmonary toxicity can develop even following a single course of bleomycin. It is vital to consider the importance of reassessment of pulmonary function after each course, as this will contribute to the early recognition of pulmonary toxicity.

Chest X-rays and chest computed tomography scans were performed on all patients to rule out other differential diagnoses, primarily lung infection and chronic lung disease. Patients were admitted to the hospital during the acute phase of their symptoms. The duration of hospitalization ranged from a minimum of three days to a maximum of twenty days, with a median value of 10.4 days. Following the conclusion of the designated period, no patients continued to experience persistent dyspnea complaints. In two cases, the disease progressed in patients who had developed pulmonary toxicity after being administered bleomycin. One of the patients entered a state of remission subsequent to the administration of first-line treatment, while the other patient achieved remission following the implementation of a second-line treatment plan. At the time of their most recent follow-up, all patients who had developed pulmonary toxicity were being monitored as treatment responders.

When considering drug-related toxicity, opportunistic lung infections, radiation damage, heart failure, lymphangitic carcinomatosis, pulmonary metastasis, and pulmonary tumor embolism should be considered in the differential diagnosis [18]. The suspected causative agent should be discontinued after preliminary diagnosis. There is no proven treatment in the literature other than glucocorticoid use and empirical supportive care [12,19]. Supportive care includes inhaled bronchodilators, supplemental oxygen, and mechanical ventilation. The degree of pulmonary toxicity determines the decision to reuse the causative drug [7,20].

Bleomycin is a significant antineoplastic agent employed in both adjuvant therapy and the metastatic stage of TC, the most prevalent solid malignancy in young males. Pulmonary damage has been observed in up to 10% cases involving the administration of bleomycin, an antitumor antibiotic [21,22,23].

In this study, a higher incidence of bleomycin-related symptomatic pulmonary toxicity (23 patients (19.49%)) was recorded than that reported in the literature. It is hypothesized that this increase can be attributed to several concomitant pathophysiological factors. The study revealed a high rate of granulocyte colony-stimulating factor (GCSF) use (12 patients (54.55%)) among patients who developed pulmonary toxicity. Retrospective records indicated that this was due to prolonged neutropenia. It was hypothesized that GCSF use may have increased the toxicity rate. The concomitant use of GCSF in bleomycin-containing treatment protocols is a risk factor for the development of bleomycin-related pulmonary damage in animal studies. However, human studies have yielded both positive and negative results. The rationale for this phenomenon, as elucidated by the extant literature, has been attributed to disparities in median age [24,25,26,27,28,29].

The primary rationale behind clinicians’ hesitancy to employ GCSF during bleomycin-containing therapy pertains to its potential for pulmonary toxicity. However, should a patient have a documented history of febrile neutropenia, the prevailing treatment approach entails the administration of GCSF on the day of bleomycin infusion [30].

Another potential explanation may be the high prevalence of a history of smoking. The study found that 42 patients (35.59%) were active smokers, and pulmonary toxicity was observed in 13 of them (31%). Most cases of pulmonary toxicity were observed in patients who smoked (13 patients [56.52%]). The pulmonary toxicity levels in smokers were found to be 5.2 times higher than those observed in non-smokers. This finding indicated that smoking-related chronic inflammation may augment the risk of drug-related pulmonary toxicity. Despite the existence of studies within the extant literature that appear to support this finding [10,31,32,33], it should be noted that there have also been studies that have failed to obtain significant results [26,34].

Drug-related lung injury is more common in elderly patients [23]. In this study, pulmonary toxicity was most common in the 15–29 age group (10 patients (43.48%), with its frequency decreasing with increasing age. The positive aspect of this result is that none of the patients who developed pulmonary toxicity died. In the literature, the mean age of patients without fatal toxicity is 33, while the mean age of patients with fatal pulmonary toxicity is 55 [35]. This result supports the absence of mortality in our study, which is attributed to the low median age at diagnosis.

Drug-related pulmonary toxicity most commonly presents symptoms within one to six months after the start of treatment. Clinical signs, symptoms, physical examination findings, and radiological findings are nonspecific. A restrictive SFT pattern and decreased DLCO may be present. It is a diagnosis of exclusion. Resolution of symptoms following discontinuation of treatment supports the diagnosis [36].

In this study, patients who developed antineoplastic-related pulmonary toxicity after differential diagnosis were identified from oncology file records. Patients with dyspnea after bleomycin administration were included in the pulmonary toxicity group. A DLCO decrease of more than 10% was observed in 66.67% of patients (n = 14), while a DLCO decrease of 10% or less was observed in 33.33% (n = 7). This result was consistent with the literature. Bleomycin-related toxicity may be accompanied by a significant decrease in DLCO. However, small changes that are not correlated with symptoms may also occur [11,37,38]. If DLCO is shown to be below 30–35% of the baseline value in drug toxicity, discontinuation of bleomycin use is recommended [39,40]. In 13% of patients (n:3), the DLCO decrease was above 30–35%, but this did not affect the follow-up surveys.

It was noted that bleomycin use was discontinued in all symptomatic patients, and they completed the planned oncological treatment. Inhaler support therapy and intravenous steroids were administered. Symptoms subsided, and after completing treatment without bleomycin, they were placed on follow-up without treatment. Nineteen patients (82.61%) had received bleomycin during the adjuvant period, while the remainder had received it during first-line treatment in the postoperative relapse phase. Age at diagnosis, performance status, histopathological subgroup, tumor size, diagnostic symptom, stage, number of adjuvant treatment cycles, and tumor marker levels were similar in the groups with and without toxicity. Therefore, the prognostic value of lymphocyte-related inflammation markers (NLR, PLR, CLR, SII, and SIRI) was interpreted more reliably. Statistical analysis results showed that lymphocyte-associated inflammation markers (NLR > 1.64, PLR > 93.92, CLR > 0.49, SII > 444.25, and SIRI > 0.66) evaluated at the time of initial diagnosis and before starting bleomycin treatment were prognostically useful for predicting pulmonary toxicity.

Radiographic findings of bleomycin-induced pulmonary toxicity are nonspecific, including ground-glass opacity, increased reticular markings, pleural effusion, focal consolidation areas, volume loss, and blunting of the costophrenic angle [7,41,42]. Chest X-ray is used to rule out differential diagnoses [43]. Restricted spirometry patterns, such as decreased forced vital capacity (FVC) and total lung capacity (TLC), are present. The clinical manifestation of this is a decrease in oxygen saturation and dyspnea at rest and during exertion due to abnormalities in gas exchange [44]. As was the case with the initial complaint of all patients who developed bleomycin-related pulmonary toxicity in this study, the most common symptom of drug-related pulmonary toxicity is dyspnea [10].

In a study conducted in patients with non-small-cell lung cancer, it was observed that the prognosis was poor in patients who developed pulmonary toxicity, with an mOS of 3.5 months [45].

Pulmonary toxicity is one of the side effects that alarm clinicians, as we do not know what will happen to these patients. So, the idea behind this study is to add to the existing knowledge on the subject. As all patients with pulmonary toxicity survived in this study, we could not calculate OS. This suggests that bleomycin-related pulmonary toxicity is not necessarily a sign of a poor prognosis, despite survival data showing that lung cancer patients who develop pulmonary toxicity have a shorter life expectancy.

BAL has frequently revealed lymphocytosis, neutrophilia, and eosinophilia in cases of pulmonary toxicity [7]. Histopathological findings include interstitial pneumonia, diffuse alveolar damage, hypersensitivity pneumonia, eosinophilic pneumonia, non-necrotizing granulomatous inflammation, pulmonary Veno-occlusive disease, and alveolar hemorrhage [12]. The most common findings in BAL suggested that lymphocyte-associated inflammation markers (NLR, PLR, LMR, CLR, SII, and SIRI) may be prognostic markers that allow us to predict bleomycin-associated pulmonary toxicity. Atypical mice are resistant to bleomycin-related lung toxicity. This result demonstrates the importance of the inflammatory process in the pathogenesis of the disease [13].

All patients scheduled to receive bleomycin should be assessed for the risk of developing drug-related pulmonary toxicity prior to treatment. SFT and DLCO should be performed prior to treatment to determine lung capacity. They should be repeated when symptoms develop, and serial measurements should be taken. The optimal test frequency has not been determined. TC patients who receive three/four courses of BEP (bleomycin, etoposide, cisplatin) should be screened for concomitant GCSF use, a concurrent radiotherapy plan to the lung parenchyma, chronic lung disease, advanced age, and a history of smoking [46]. In addition, the use of lymphocyte-related inflammation markers (NLR, PLR, LMR, CLR, SII, and SIRI) in follow-up is thought to enable clinicians to predict patients with a high likelihood of developing pulmonary toxicity. Serial follow-up of prognostic tests is particularly important for patients with high risk factors. Reassessment before each treatment cycle is recommended.

Following the first publications suggesting that a decrease in DLCO is the earliest indicator of pulmonary toxicity leading to bleomycin discontinuation [47,48], studies emerged suggesting that SFT and DLCO are neither specific nor sensitive [10,49,50,51]. Finally, in a randomized controlled phase 3 trial, it was recommended that SFT be performed only in symptomatic patients and that computed tomography of the chest be performed early if deemed necessary [52]. In this patient group, where there is still no specific standardization, it is thought that the use of lymphocyte-related inflammation markers (NLR, PLR, CLR, SII, and SIRI), which can be calculated from the results of routine pre-treatment blood counts and biochemical tests, would be beneficial.

The process of bleomycin-associated pulmonary damage is characterized by a self-limiting course of events, which progresses through the initial phase of inflammation (0–7 days), the fibrogenic phase (7–21 days), and the resolution phase (42–56 days) [53].

Intratracheal administration of bleomycin in mice resulted in pulmonary fibrosis, and inflammatory cell infiltration in the alveolar walls became evident on the seventh day. During this period, an increase in neutrophil count was observed initially, and lymphocyte count peaked on the 21st day. Tumor necrosis factor-α (TNF-α) and transforming growth factor-β1 (TGF-β1) levels increased progressively until the 21st day after bleomycin administration. This result is hypothesized to elucidate the correlation between the inflammatory response and the incidence of pulmonary toxicity [54].

This result supports the conclusion that the lymphocyte-associated inflammation markers evaluated in our study are prognostic. However, we believe that analyzing the correlation between lymphocyte-associated inflammation markers in BAL and peripheral blood in future planned clinical studies will contribute to this field of research. The prognostic power of the NLR ratio in peripheral blood has been demonstrated in various cancer groups. However, no clinical study other than animal studies investigating bleomycin-associated pulmonary toxicity has examined the correlation between the NLR ratio in BAL fluid and the NLR ratio in peripheral blood. Unlike other body fluids, BAL fluid is in direct contact with pulmonary tissue. Therefore, we believe that the results of lymphocyte-related inflammation markers calculated from BAL fluid may have higher prognostic power than those calculated from peripheral blood.

Although neutrophil increase was evaluated in animal experiments in our literature review, we observed that the studies included NLR and other markers in this study provided very limited results. The NLR ratio in peripheral blood and BAL fluid has been investigated in lung cancer patients in the literature. Both groups were found to be prognostic, and this study was the first to show that NLR in BAL fluid is poor prognostic. Cut-off values differed, and no correlation was observed between NLR calculated from peripheral blood and BAL fluid [55].

Increases in IL-1β, IL-18, CTGF, and TNF-α first cause neutrophil and macrophage inflammation, followed by an increase in extracellular matrix and fibroblasts. Neutrophils are the first immune cells to form the inflammatory response through chemotaxis and extravasation in the target tissue. The inflammatory response initiates the fibrotic process. Measuring IL-1β, IL-18, CTGF, and TNF-α in BAL fluid along with pulmonary function tests may enable early diagnosis of patients who will develop fibrosis. On the other hand, it is still unclear whether neutrophils increase the fibrotic process or whether they increase repaired damaged pulmonary tissue [56].

Alveolar macrophages located in the pulmonary airways are responsible for the homeostasis of lung tissue. MicroRNAs (miRNAs), which are small non-coding RNA clusters, are thought to play an important role in immune cell development by providing epigenetic regulation in this process. It has been shown that miRNAs play a critical role in regulating alveolar macrophages and the inflammatory response in both healthy and diseased states [57].

In recent years, transcriptomic-profiling-based studies have been published in the literature to elucidate the molecular mechanism of bleomycin-associated pulmonary toxicity. Examples include spatial transcriptomics, single-cell RNA-seq, and RNA-seq. These studies have led to the identification of new biomarkers that reveal changes in gene expression during the early stages of pulmonary toxicity. In an animal study described in the literature, where single-cell RNA sequencing was performed after bleomycin administration, an increase in endothelial cells characterized by Cxcl12 was observed in bleomycin-induced pulmonary fibrosis. As a result of this increase, chemokine activity, angiogenesis regulation, and collagen binding increased, and consequently, the profibrotic process began. This process, which begins with the activation of proinflammatory pathways and extracellular matrix-receptor activation, ends in fibrosis [58].

Single-cell RNA sequencing was used in another animal study. Keratin-8-positive progenitor cells were identified in a bleomycin induced pulmonary fibrosis model. It was found that the function of this transient cell population is to maintain the balance between fibrosis and tissue regeneration. It has been suggested that these cells could be used as an early indicator of toxicity [59].

The complexity of the chemokine system has been highlighted in a mouse model from animal studies investigating bleomycin induced pulmonary fibrosis. The chemokine axes (CCL19/CCL21-CCR7, CCL8-CCR2, CXCL9-CXCR3, CCL20-CCR6, and CCL3/CCL4/CCL5-CCR5) are thought to play a central role in fibrogenesis. Oxidative damage initiated by bleomycin use causes DNA and RNA breaks. This is followed by the release of chemokines, growth factors, and metalloproteinases. The inflammatory response induces fibrosis [60].

The increase in all immune cells, particularly lymphocytes, is involved in the inflammatory response, and the increase in prognostic markers determined using immune cell levels are therefore considered prognostic for the development of pulmonary toxicity. The use of lymphocyte-related inflammation markers (NLR, PLR, CLR, SII, and SIRI), which we emphasized in the results of our study, will therefore be useful. Demonstrating the increase in these chemokines in peripheral blood samples is thought to be promising for the use of biomarkers that can be used for early diagnosis. These findings suggest that, in addition to clinical and functional pulmonary assessments, transcriptomic analyses may enable non-invasive early diagnosis of bleomycin-related pulmonary toxicity and prediction of its severity. In the future, we believe that integrating transcriptomic data analysis with results obtained from clinical, laboratory, and imaging methods will lead to the emergence of clinically applicable biomarkers.

The most important limitations of our study are that it is a single-center experience and that the data were obtained from retrospective medical records. Due to the limited number of events in our study, only a few variables could be included in the multivariate analysis. Consequently, some potentially inflammatory markers were excluded from the final model. This is a common limitation in studies evaluating inflammatory markers. Future prospective studies will guide the prognostic value of lymphocyte-related inflammation markers (NLR, PLR, CLR, SII, and SIRI) in patients scheduled for bleomycin.

The exclusion criteria for this study included having a second malignancy, a history of chronic disease (diabetes mellitus, chronic renal failure, chronic rheumatic disease, chronic lung disease), obesity, and a history of pulmonary surgery/radiotherapy. All these exclusion criteria are variables that can increase inflammation markers. By excluding these patients, confounders that could affect the results of the inflammation scores were prevented. The absence of such an exclusion criterion in studies in the literature adds value to the results of this study.

## 5. Conclusions

The diagnosis of antineoplastic-agent-related pulmonary toxicity is a detailed exclusion diagnosis. When drug toxicity is considered, the treatment agent should be discontinued. The reduction in symptoms following discontinuation of treatment supports the diagnosis. There is no proven treatment other than glucocorticoid use and empirical supportive care. The degree of pulmonary toxicity determines the decision to restart the drug. Lymphocyte-associated inflammation markers (NLR, PLR, CLR, SII, and SIRI) are prognostic in the development of drug-related pulmonary toxicity in patients with testicular cancer receiving bleomycin therapy.

In the future, advanced transcriptomic profiling approaches may enable a detailed understanding of the molecular mechanisms that could predict bleomycin-related pulmonary toxicity at an early stage. Researchers are interested in whether the widespread and comprehensive use of this technology could lead to the identification of specific biomarkers and the development of preventive strategies. 

## Figures and Tables

**Figure 1 jcm-14-07926-f001:**
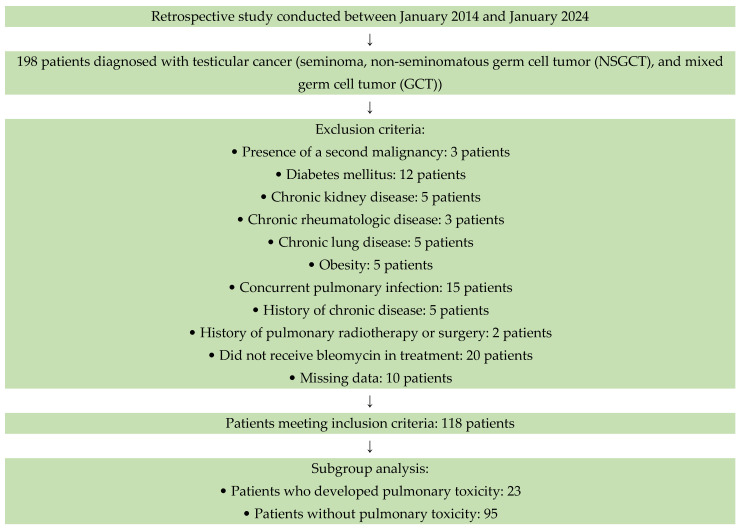
Flow diagram of patient selection for the retrospective study conducted between January 2014 and January 2024.

**Figure 2 jcm-14-07926-f002:**
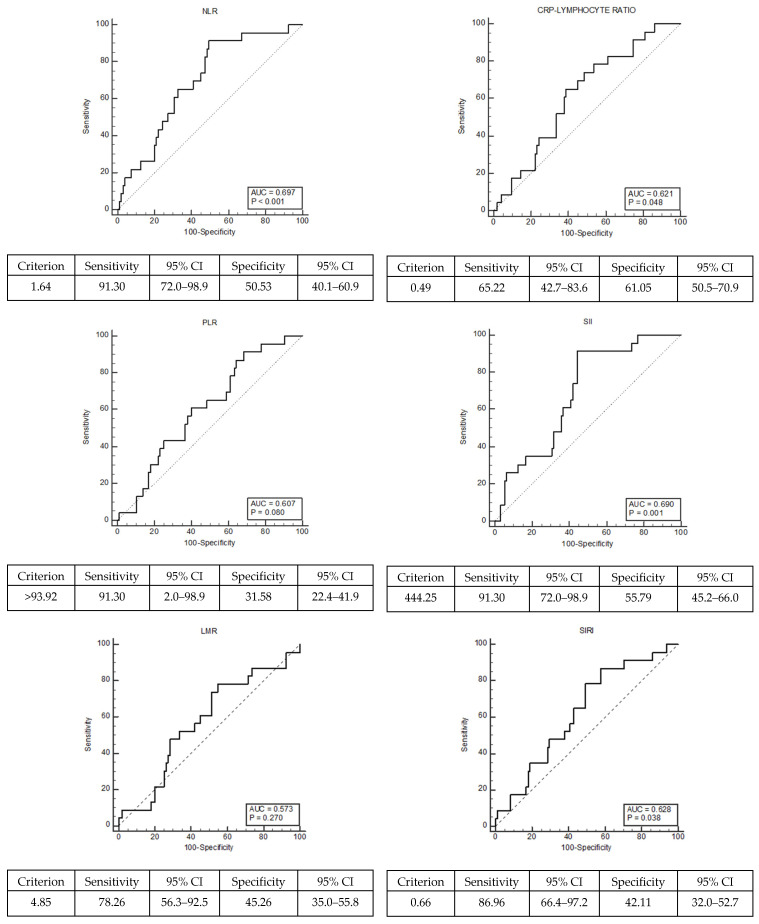
Roc curve analysis of lymphocyte-related inflammation markers (neutrophil/lymphocyte ratio (NLR); platelet/lymphocyte ratio (PLR); lymphocyte/monocyte ratio (LMR); CRP/lymphocyte ratio (CLR); systemic immune inflammation score (SII); systemic inflammation response index (SIRI)).

**Figure 3 jcm-14-07926-f003:**
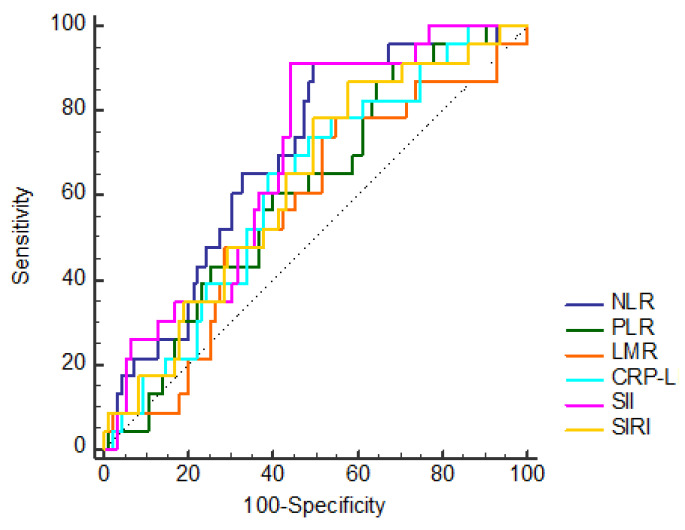
Roc curve analysis of lymphocyte-related inflammation markers (neutrophil/lymphocyte ratio (NLR); platelet/lymphocyte ratio (PLR); lymphocyte/monocyte ratio (LMR); CRP/lymphocyte ratio (CRP-L); systemic immune inflammation score (SII); systemic inflammation response index (SIRI)).

**Table 1 jcm-14-07926-t001:** Clinical and demographic characteristics of testicular cancer patients receiving bleomycin in oncological treatment.

Variables		N (%)
Age at diagnosis (median ± SS)	32.19 ± 9.62	30.79 (15.73–65.33)
Age at diagnosis	15–29 years	53 (44.92)
	30–39 years	43 (36.44)
	40–49 years	15 (12.71)
	≥50 years	7 (5.93)
Eastern Cooperative Oncology Group (ECOG) performance status	0	112 (94.92)
	1	3 (2.54)
	2	1 (0.85)
	3	2 (1.69)
Histopathology	Seminoma	43 (36.44)
	Embryonal carcinoma	20 (16.95)
	Mixed germ cell carcinoma	54 (45.76)
Epididymal invasion	No	112 (94.92)
	Yes	6 (5.08)
Tunica albuginea invasion	No	101 (85.59)
	Yes	17 (14.41)
Tunica vaginalis invasion	No	112 (94.92)
	Yes	6 (5.08)
Lymphovascular invasion	No	52 (44.07)
	Yes	66 (55.93)
Tumor localization	Right	56 (47.46)
	Left	59 (50)
	Bilateral	3 (2.54)
Diagnostic symptom	Mass	53 (44.92)
	Swelling	35 (29.66)
	Pain	30 (25.42)
Stage	Stage 1	69 (58.47)
	Stage 2	29 (24.58)
	Stage 3	20 (16.95)
Tumor size	42.61 ± 24.7	35 (6–120)
Radiotherapy	No	113 (95.76)
	Yes	5 (4.24)
Surgery	No	4 (3.39)
	Yes	114 (96.61)
Retroperitoneal lymph node dissection	No	86 (72.88)
	Yes	32 (27.12)
Adjuvant therapy	No	19 (16.1)
	Yes	99 (83.9)
Number of adjuvant cycles	2.96 ± 0.94	3 (1–6)
Pulmonary toxicity	No	95 (80.51)
	Yes	23 (19.49)
Decrease in DLCO in patients with pulmonary toxicity	≤10%	7 (33.33)
	>10%	14 (66.67)
Use of GCSF in patients developing pulmonary toxicity	No	10 (45.45)
	Yes	12 (54.55)
Progression	No	93 (78.81)
	Yes	25 (21.19)
Living situation	Alive	107 (90.68)
	Exitus	11 (9.32)
Median overall survival (mOS)	159.86 ± 4.34 (min/max:151.346–168.364)	
NLR	≤1.64	50 (42.37)
	>1.64	68 (57.63)
PLR	≤93.92	32 (27.12)
	>93.92	86 (72.88)
LMR	≤4.85	70 (59.32)
	>4.85	48 (40.68)
CLR	≤0.49	66 (55.93)
	>0.49	52 (44.07)
SII	≤444.25	55 (46.61)
	>444.25	63 (53.39)
SIRI	≤0.66	43 (36.44)
	>0.66	75 (63.56)
Median forced vital capacity (FVC)	85.96 ± 7.41	87 (73–105)
Median carbon monoxide diffusion capacity (DLCO)	68.18 ± 13.44	70 (34–88)

SS: Sum of Squares; NLR: neutrophil/lymphocyte ratio; PLR: platelet/lymphocyte ratio; LMR: lymphocyte/monocyte ratio; CLR: CRP/lymphocyte ratio; SII: systemic immune inflammation score; SIRI: systemic inflammation response index.

**Table 2 jcm-14-07926-t002:** Comparison of clinical and demographic characteristics according to the severity of bleomycin-related pulmonary toxicity.

Variables		Pulmonary Toxicity		*p*
		No	Yes	
		N (%)	N (%)	
Age at diagnosis	15–29 years	43 (45.26)	10 (43.48)	0.880
	30–39 years	35 (36.84)	8 (34.78)	
	40–49 years	12 (12.63)	3 (13.04)	
	≥50 years	5 (5.26)	2 (8.7)	
Smoking history	No	66 (69.47)	10 (43.48)	0.028 *
	Yes	29 (30.53)	13 (56.52)	
Eastern Cooperative Oncology Group (ECOG) performance status	0	89 (93.68)	23 (100)	1.000
	1	3 (3.16)	0 (0)	
	2	1 (1.05)	0 (0)	
	3	2 (2.11)	0 (0)	
Histopathology	Seminoma	34 (35.79)	9 (39.13)	0.716
	Embryonal carcinoma	15 (15.79)	5 (21.74)	
	Mixed germ cell carcinoma	45 (47.37)	9 (39.13)	
Epididymal invasion	No	90 (94.74)	22 (95.65)	1.000
	Yes	5 (5.26)	1 (4.35)	
Tunica albuginea invasion	No	82 (86.32)	19 (82.61)	0.741
	Yes	13 (13.68)	4 (17.39)	
Tunica vaginalis invasion	No	89 (93.68)	23 (100)	0.596
	Yes	6 (6.32)	0 (0)	
Lymphovascular invasion	No	43 (45.26)	9 (39.13)	0.646
	Yes	52 (54.74)	14 (60.87)	
Tumor localization	Right	46 (48.42)	10 (43.48)	0.818
	Left	46 (48.42)	13 (56.52)	
	Bilateral	3 (3.16)	0 (0)	
Diagnostic symptom	Mass	40 (42.11)	13 (56.52)	0.362
	Swelling	31 (32.63)	4 (17.39)	
	Pain	24 (25.26)	6 (26.09)	
Stage	Stage 1	56 (58.95)	13 (56.52)	0.381
	Stage 2	25 (26.32)	4 (17.39)	
	Stage 3	14 (14.74)	6 (26.09)	
Radiotherapy	No	90 (94.74)	23 (100)	0.582
	Yes	5 (5.26)	0 (0)	
Surgery	No	4 (4.21)	0 (0)	1.000
	Yes	91 (95.79)	23 (100)	
Retroperitoneal lymph node dissection	No	69 (72.63)	17 (73.91)	1.000
	Yes	26 (27.37)	6 (26.09)	
Adjuvant therapy	No	15 (15.79)	4 (17.39)	1.000
	Yes	80 (84.21)	19 (82.61)	
NLR	≤1.64	48 (50.53)	2 (8.7)	0.001 *
	>1.64	47 (49.47)	21 (91.3)	
PLR	≤93.92	30 (31.58)	2 (8.7)	0.035 *
	>93.92	65 (68.42)	21 (91.3)	
LMR	≤4.85	52 (54.74)	18 (78.26)	0.057
	>4.85	43 (45.26)	5 (21.74)	
CLR	≤0.49	58 (61.05)	8 (34.78)	0.034 *
	>0.49	37 (38.95)	15 (65.22)	
SII	≤444.25	53 (55.79)	2 (8.7)	0.001 *
	>444.25	42 (44.21)	21 (91.3)	
SIRI	≤0.66	40 (42.11)	3 (13.04)	0.014 *
	>0.66	55 (57.89)	20 (86.96)	

NLR: neutrophil/lymphocyte ratio; PLR: platelet/lymphocyte ratio; LMR: lymphocyte/monocyte ratio; CLR: CRP/lymphocyte ratio; SII: systemic immune inflammation score; SIRI: systemic inflammation response index; Fisher–Freeman–Halton Exact Test. Statistically significant results (*p* < 0.05) are indicated with a (*) sign next to the *p* value.

**Table 3 jcm-14-07926-t003:** Univariate and multivariate logistic regression for pulmonary toxicity.

Variables		Univariate Analysis		Multivariate Analysis	
		OR (95% CI)	*p*	OR (95% CI)	*p*
Age at diagnosis	15–29 years	1 (reference)	0.941		
	30–39 years	0.983 (0.35–2.756)	0.974		
	40–49 years	1.075 (0.255–4.538)	0.922		
	≥50 years	1.72 (0.291–10.182)	0.550		
Smoking history	Yes	2.959 (1.164–7.52)	0.023 *	5.23 (1.536–17.814)	0.008 *
Histopathology	Seminoma	1 (reference)	0.872		
	Embryonal carcinoma	1.259 (0.361–4.398)	0.718		
	Mixed germ cell carcinoma	0.756 (0.271–2.107)	0.592		
Lymphovascular invasion	No	1 (reference)			
	Yes	1.286 (0.508–3.259)	0.596		
Tumor localization	Right	1 (reference)	0.855		
	Left	1.3 (0.518–3.263)	0.576		
Diagnostic symptom	Mass	1 (reference)	0.329		
	Swelling	0.397 (0.118–1.338)	0.136		
	Pain	0.769 (0.258–2.292)	0.638		
Stage	Stage 1	1 (reference)	0.374		
	Stage 2	0.689 (0.204–2.325)	0.549		
	Stage 3	1.846 (0.596–5.72)	0.288		
Retroperitoneal lymph node dissection	No	1 (reference)			
	Yes	0.937 (0.333–2.635)	0.901		
NLR	≤1.64	1 (reference)			
	>1.64	10.723 (2.38–48.306)	0.002 *		
PLR	≤93.92	1 (reference)			
	>93.92	4.846 (1.067–22.015)	0.041 *		
LMR	≤4.85	1 (reference)			
	>4.85	0.336 (0.115–0.979)	0.046*		
CLR	≤0.49	1 (reference)			
	>0.49	2.939 (1.134–7.615)	0.026 *		
SII	≤444.25	1 (reference)			
	>444.25	13.25 (2.939–59.731)	0.001 *		
SIRI	≤0.66	1 (reference)			
	>0.66	4.848 (1.348–17.439)	0.016 *		

NLR: neutrophil/lymphocyte ratio; PLR: platelet/lymphocyte ratio; LMR: lymphocyte/monocyte ratio; CLR: CRP/lymphocyte ratio; SII: systemic immune inflammation score; SIRI: systemic inflammation response index; statistically significant results (*p* < 0.05) are indicated with a (*) sign next to the *p* value.

## Data Availability

The data underlying this article will be shared at reasonable request to the corresponding author.

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
