# Peer review of "Lymphocyte-Associated Inflammation Markers Predict Bleomycin-Induced Pulmonary Toxicity in Testicular Cancer"

_jcm, 2025, doi:10.3390/jcm14227926_

Round 1

Reviewer 1 Report

Comments and Suggestions for Authors

An interesting manuscript that show a relationship between bleomycin treatment and pulmonary toxicity. Despite the numbers of patient is low, some interesting finding are presented.

The manuscript is well written and structured and some minor changes are required.

1) All the patients were included in the study. No inclusion or exclusion criteria are indicated. This should be clarified.

2) In patients there was any anlysis indicating that some of them  showed renal impairment)

3) Discussion section require improvement; novel approach to evaluate BPT are under research and must be included in this section. For example, transcriptomic profile that allow detect novel biomarkers and probably can be a new approach to detect BPT.

After these minor changes will be incorporated, manuscript is ready for a new review. 

Author Response

We sincerely appreciate your time and effort in reviewing our manuscript. Detailed responses to the reviewers’ comments are provided below, and all corresponding revisions and corrections have been highlighted (or shown in track changes) in the revised submission

Open Review

Quality of English Language

( ) The English could be improved to more clearly express the research.
(x) The English is fine and does not require any improvement.

Yes

Can be improved

Must be improved

Not applicable

Does the introduction provide sufficient background and include all relevant references?

(x)

( )

( )

( )

Is the research design appropriate?

(x)

( )

( )

( )

Are the methods adequately described?

( )

(x)

( )

( )

Are the results clearly presented?

(x)

( )

( )

( )

Are the conclusions supported by the results?

(x)

( )

( )

( )

Are all figures and tables clear and well-presented?

(x)

( )

( )

( )

Dear Reviewer,

We appreciate your emphasis on the difficulty and complexity of diagnosing pulmonary toxicity caused by antineoplastic agents, which relies on a comprehensive exclusion process. As we mentioned in the Discussion section, considering the difficulty of this diagnosis, we excluded all patients who could be included in the differential diagnosis in our study, thereby increasing the validity and power of our analyses. We have presented the inclusion and exclusion criteria for the study in detail in the Materials and Methods section.

                We also thank you very much for your constructive and positive feedback. During our literature review, we gained a better understanding of the importance of new transcriptomic-based approaches in biomarker identification and developed our Discussion and Materials and Methods sections accordingly. We thank you again for your valuable contributions.

Best regards.

  • Comments 1: All the patients were included in the study. No inclusion or exclusion criteria are indicated. This should be clarified.

This study, as a single-center experience, reviewed the data of 198 testicular cancer patients diagnosed with TC (seminoma, NSGCT and mixed germ cell tumor) who presented to the Medical Oncology Clinic between January 2014 and January 2024. A total of 118 TC patients who received bleomycin at any stage of treatment and met the inclusion and exclusion criteria were included. To improve clarity, a patient enrollment flowchart has been added to the revised version of the manuscript.

(Page:3 , Line 98-102)

Inclusion and exclusion criteria of the study

Inclusion criteria:

Patients aged 18 years or older with a diagnosis of testicular cancer who were followed up at the oncology clinic during the specified period (January 2014-January 2024), received bleomycin in the adjuvant or first-line treatment, and met the exclusion criteria were consecutively enrolled in the study

Exclusion criteria:

Patients with any of the following conditions were excluded from the study: the presence of a second malignancy diagnosis, diabetes mellitus, chronic renal failure, chronic rheumatic disease, or chronic lung disease. Patients with obesity, concomitant pulmonary infection, or a history of pulmonary radiotherapy or surgery were also excluded. In addition, individuals who had not received bleomycin as part of their treatment regimen were not included in the study

Page:3, 4 , Line : 115-125

Retrospective study conducted between January 2014 and January 2024

198 patients diagnosed with testicular cancer (seminoma, non-seminomatous germ cell tumor (NSGCT), and mixed germ cell tumor (GCT))

Exclusion criteria:
• Presence of a second malignancy: 3 patients
• Diabetes mellitus: 12 patients
• Chronic kidney disease: 5 patients
• Chronic rheumatologic disease: 3 patients
• Chronic lung disease: 5 patients
• Obesity: 5 patients
• Concurrent pulmonary infection: 15 patients
• History of chronic disease: 5 patients
• History of pulmonary radiotherapy or surgery: 2 patients
• Did not receive bleomycin in treatment: 20 patients
• Missing data: 10 patients

Patients meeting inclusion criteria: 118 patients

Subgroup analysis:
• Patients who developed pulmonary toxicity: 23
• Patients without pulmonary toxicity: 95

Figure 1. Flow diagram of patient selection for the retrospective study conducted between January 2014 and January 2024.

 (page:4 line:125-150)

  • Comments 2: In patients there was any anlysis indicating that some of them  showed renal impairment

Patients with chronic kidney failure were not included in the study. Patients were excluded based on their creatinine levels and glomerular filtration rate.

  • Comment 3: Discussion section require improvement; novel approach to evaluate BPT are under research and must be included in this section. For example, transcriptomic profile that allow detect novel biomarkers and probably can be a new approach to detect BPT.

We appreciate the reviewer’s valuable comment regarding the improvement of the Discussion section. In accordance with this suggestion, the Discussion has been revised and expanded. New approaches for evaluating BPT have been explored and incorporated into the section. In particular, transcriptomic profiling which allows the identification of novel biomarkers and may represent a promising new approach for detecting BPT has been discussed based on a focused review of the relevant literature.

 (Discussion section ; page: ,line: )

We sincerely thank the reviewer for the positive and encouraging comments. We truly appreciate the time and effort dedicated to reviewing our manuscript. All suggested revisions have been carefully addressed, and we believe that these improvements have strengthened the overall quality and contribution of our study.

Reviewer 2 Report

Comments and Suggestions for Authors

In this study, Melek Özdemir et al. investigated the predictive value of peripheral blood lymphocyte-related inflammatory indicators (NLR, PLR, LMR, CLR, SII, SIRI) for symptomatic pulmonary toxicity in patients with testicular cancer (TC) after receiving a bleomycin-based treatment regimen. The authors found: â‘  19.5% of the patients experienced symptomatic pulmonary toxicity; â‘¡ smokers had a 5.2-fold higher risk of pulmonary toxicity; â‘¢ all six indicators were independently associated with pulmonary toxicity when they were higher than the established cutoff values before treatment. The conclusion is that these routinely accessible inflammatory indicators can be used as predictive tools for bleomycin-induced pulmonary toxicity. This is an interesting study, but the following issues still need to be addressed:

  1. This study did not adopt a retrospective single-center design, which leads to unavoidable confounding, selection bias, and information loss. The article did not explain or handle the missing data (such as smoking quantity, number of chemotherapy cycles, concomitant medications, imaging details).
  2. Among 118 cases, only 23 events occurred. In the univariate analysis, 10 variables emerged, and the multivariate model included 6 inflammatory indicators + smoking history. This is prone to overfitting. It is necessary to compress the variables according to the "number of events/variables ≥ 10" principle and report the model's PS efficacy or post hoc efficacy calculation.
  3. The diagnosis was based solely on "symptoms + DLCO decline > 10%". No imaging blind review, BAL/HRCT proportion, or specific procedures for excluding infection/transfer were provided. It is recommended to supplement the STROBE flowchart and explain the consensus used for diagnosis (such as CIRCO or FDA adverse event standards).
  4. In the multivariate analysis of Table 3, the OR confidence interval was too wide (such as for NLR > 1.64, the 95% CI of OR was 3.81–37.3), suggesting insufficient sample size; the VIF collinearity test should be reported.
  5. Although all patients with pulmonary toxicity survived, the recurrence/progression/length of hospitalization/pulmonary function recovery time should be reported to compensate for the limitation of OS calculation.
  6. Why does the "lymphocyte-dominated" BAL result necessarily correspond to an increase in peripheral blood NLR and other indicators? It is necessary to cite basic or animal studies to explain this.
  7. There are multiple spelling and grammar errors in the writing (such as "toksisity" "rite rion"), which need to be proofread by a native speaker.
  8. The reference list format needs further optimization.

I am very pleased to read your paper "Lymphocyte-associated inflammation markers predict bleomycin-induced pulmonary toxicity in testicular cancer". I thank you for your work. I encourage you to address the issues mentioned as they can enhance the contribution of this manuscript to the field of this research.

Author Response

We sincerely appreciate your time and effort in reviewing our manuscript. Detailed responses to the reviewers’ comments are provided below, and all corresponding revisions and corrections have been highlighted (or shown in track changes) in the revised submission.

Reviewer 2:

Quality of English Language

(x) The English could be improved to more clearly express the research.
( ) The English is fine and does not require any improvement.

Yes

Can be improved

Must be improved

Not applicable

Does the introduction provide sufficient background and include all relevant references?

(x)

( )

( )

( )

Is the research design appropriate?

( )

(x)

( )

( )

Are the methods adequately described?

( )

(x)

( )

( )

Are the results clearly presented?

( )

(x)

( )

( )

Are the conclusions supported by the results?

(x)

( )

( )

( )

Are all figures and tables clear and well-presented?

( )

(x)

( )

( )

In this study, Melek Özdemir et al. investigated the predictive value of peripheral blood lymphocyte-related inflammatory indicators (NLR, PLR, LMR, CLR, SII, SIRI) for symptomatic pulmonary toxicity in patients with testicular cancer (TC) after receiving a bleomycin-based treatment regimen. The authors found: â‘  19.5% of the patients experienced symptomatic pulmonary toxicity; â‘¡ smokers had a 5.2-fold higher risk of pulmonary toxicity; â‘¢ all six indicators were independently associated with pulmonary toxicity when they were higher than the established cutoff values before treatment. The conclusion is that these routinely accessible inflammatory indicators can be used as predictive tools for bleomycin-induced pulmonary toxicity. This is an interesting study, but the following issues still need to be addressed:

  1. Comments: This study did not adopt a retrospective single-center design, which leads to unavoidable confounding, selection bias, and information loss. The article did not explain or handle the missing data (such as smoking quantity, number of chemotherapy cycles, concomitant medications, imaging details).

Response 1: (Results; page:5, 6;line: 182-184; 198-221)

We appreciate the reviewer’s valuable feedback. Since our study was designed retrospectively, smoking history was recorded only as “present” or “absent”; therefore, quantitative information such as the number of cigarettes could not be provided. Patients with chronic diseases were excluded from the study, so there were no participants using long-term medications. In cases where pulmonary symptoms developed, management included the use of inhaled corticosteroids, inhaled bronchodilators, intravenous corticosteroids, and intermittent oxygen therapy as clinically indicated.

Regarding chemotherapy, patients received between one and three cycles of bleomycin-containing regimens, with a mean of 2.7 cycles. Four patients developed symptomatic bleomycin-induced pulmonary toxicity after only one cycle, while 19 patients had a history of three cycles. Although pulmonary toxicity occurred more frequently after three cycles, our findings indicate that it may also develop even after a single cycle. Regular reassessment of pulmonary function after each treatment cycle may facilitate earlier detection of pulmonary toxicity.

All patients underwent chest X-ray and thoracic computed tomography to exclude other possible causes, such as pulmonary infections and chronic lung diseases. During the acute phase of their symptoms, all patients were hospitalized for a minimum of three days and a maximum of ten days (median: 7.4 days). None of the patients had persistent dyspnea following discharge. Among those who developed bleomycin-related pulmonary toxicity, disease progression was observed in two patients. One patient achieved remission with first-line therapy, while the other entered remission after a second-line treatment plan was implemented. At the time of the latest follow-up, all patients who had experienced pulmonary toxicity remained in remission and were under clinical surveillance.

  1. Comments: Among 118 cases, only 23 events occurred. In the univariate analysis, 10 variables emerged, and the multivariate model included 6 inflammatory indicators + smoking history. This is prone to overfitting. It is necessary to compress the variables according to the "number of events/variables ≥ 10" principle and report the model's PS efficacy or post hoc efficacy calculation.

Response 2: (Result; limitation; page:9, line:247)

We sincerely thank the reviewer for the insightful comment regarding the “number of events/variables ≥ 10” principle. We fully agree with this point. Accordingly, since there were 23 events in our dataset, only up to 2 variables could be included in the multivariate model to satisfy this criterion. Therefore, we included smoking in the multivariate analysis, and the results are presented in Table 3. Model fit indices are also provided, and the relevant sentences have been added in the manuscript. As NLR, PLR, and other inflammatory mediators were not included in the final model, VIF analysis was not performed.

  1. Comments: The diagnosis was based solely on "symptoms + DLCO decline > 10%". No imaging blind review, BAL/HRCT proportion, or specific procedures for excluding infection/transfer were provided. It is recommended to supplement the STROBE flowchart and explain the consensus used for diagnosis (such as CIRCO or FDA adverse event standards).

Response 3: (result; page:3, 4, line:115-150)

We appreciate your valuable assessment regarding diagnostic criteria and methodological clarity. In our study, the diagnosis of bleomycin-associated pulmonary toxicity (BIPT) was based on interstitial pneumonia-compatible radiological findings, accompanied by new or worsening respiratory symptoms and a DLCO decline of more than 10% from baseline. Blood culture, sputum culture, complete blood count, CRP, sedimentation, procalcitonin, and biochemical tests were performed to rule out acute lung infection. No infection parameters were detected in any of these results, and the patients were thus included in the study. No patients showing evidence of infection in these results were included in the study.

Chest CT and chest X-rays were performed in all patients to rule out differential diagnoses such as infection, tumor progression, and chronic lung disease. The images were evaluated by a radiologist experienced in chest radiology. Although official CIRCO or FDA criteria were not applied, our diagnostic approach is consistent with the definition of bleomycin-induced pulmonary toxicity described in the literature. Diagnostic criteria and exclusion procedures are described in detail, and a strobe-compatible flow chart is provided to illustrate patient selection and the diagnostic process.

In line with the referee's suggestion, we clearly stated in the revised text that the exclusion of differential diagnoses in our study was based on previous studies in the literature [11]. In particular, the study by Yerushalmi and colleagues prospectively evaluating pulmonary function decline in patients receiving chemotherapy (Annals of Oncology, 2009;20:437–440; doi:10.1093/annonc/mdn652*) is one of the key references in this regard. In accordance with the criteria defined in this and similar studies, differential diagnoses such as infection, tumor progression, and chronic lung diseases have been systematically excluded.

Retrospective study conducted between January 2014 and January 2024

198 patients diagnosed with testicular cancer (seminoma, non-seminomatous germ cell tumor (NSGCT), and mixed germ cell tumor (GCT))

Exclusion criteria:
• Presence of a second malignancy: 3 patients
• Diabetes mellitus: 12 patients
• Chronic kidney disease: 5 patients
• Chronic rheumatologic disease: 3 patients
• Chronic lung disease: 5 patients
• Obesity: 5 patients
• Concurrent pulmonary infection: 15 patients
• History of chronic disease: 5 patients
• History of pulmonary radiotherapy or surgery: 2 patients
• Did not receive bleomycin in treatment: 20 patients
• Missing data: 10 patients

Patients meeting inclusion criteria: 118 patients

Subgroup analysis:
• Patients who developed pulmonary toxicity: 23
• Patients without pulmonary toxicity: 95

Figure 1. Flow diagram of patient selection for the retrospective study conducted between January 2014 and January 2024.

  1. Comments: In the multivariate analysis of Table 3, the OR confidence interval was too wide (such as for NLR > 1.64, the 95% CI of OR was 3.81–37.3), suggesting insufficient sample size; the VIF collinearity test should be reported.

Response 4: (results ; page3,4,line: )

Univariate and multivariate logistic regression analyses were performed for variables thought to predict pulmonary toxicity. Univariate analysis found that smoking and lymphocyte-related inflammation markers (NLR, PLR, LMR, CLR, SII, and SIRI) were statistically significant prognostic factors for pulmonary toxicity. Due to the limited number of events in our study, only a few variables could be included in the multivariate analysis. Consequently, some potentially inflammatory markers were excluded from the final model. This is a common limitation in studies evaluating inflammatory markers. Future prospective studies will guide the prognostic value of lymphocyte-related inflammation markers (NLR, PLR, CLR, SII, and SIRI) in patients scheduled for bleomycin. Smoking was confirmed as an independent predictor in the multivariate model. Other inflammatory mediators, including NLR and PLR, were excluded from the final model due to the principle of having at least 10 events per variable. These findings emphasise the significance of smoking as a determinant, while acknowledging the limitations of assessing additional biomarkers. Since only smoking was included in the final multivariate model, VIF analysis was not applicable and thus was not performed

  1. Comments: Although all patients with pulmonary toxicity survived, the recurrence/progression/length of hospitalization/pulmonary function recovery time should be reported to compensate for the limitation of OS calculation.

Response 5: (page, paragraf,line)

We thank the reviewer for this valuable suggestion. In accordance with the comment, data regarding recurrence, disease progression, length of hospitalization, and pulmonary function recovery time have now been analyzed and included in the revised manuscript. Although all patients with pulmonary toxicity survived, these additional clinical outcomes have been presented to provide a more comprehensive assessment of the patient course beyond overall survival. The relevant results have been added to the Results section and discussed in the Discussion part accordingly.

Regarding chemotherapy, patients received between one and three cycles of bleomycin-containing regimens, with a mean of 2.7 cycles. Four patients developed symptomatic bleomycin-induced pulmonary toxicity after only one cycle, while 19 patients had a history of three cycles. Although pulmonary toxicity occurred more frequently after three cycles, our findings indicate that it may also develop even after a single cycle. Regular reassessment of pulmonary function after each treatment cycle may facilitate earlier detection of pulmonary toxicity.

All patients underwent chest X-ray and thoracic computed tomography to exclude other possible causes, such as pulmonary infections and chronic lung diseases. During the acute phase of their symptoms, all patients were hospitalized for a minimum of three days and a maximum of ten days (median: 7.4 days). None of the patients had persistent dyspnea following discharge. Among those who developed bleomycin-related pulmonary toxicity, disease progression was observed in two patients. One patient achieved remission with first-line therapy, while the other entered remission after a second-line treatment plan was implemented. At the time of the latest follow-up, all patients who had experienced pulmonary toxicity remained in remission and were under clinical surveillance.

  1. Comments: Why does the "lymphocyte-dominated" BAL result necessarily correspond to an increase in peripheral blood NLR and other indicators? It is necessary to cite basic or animal studies to explain this.

Response 6: We appreciate the reviewer’s insightful comment regarding the possible relationship between a lymphocyte-predominant BAL finding and elevated peripheral blood NLR and other inflammatory markers. Although such a BAL pattern may indicate a localized inflammatory response, it could still be associated with systemic immune activation reflected by increased NLR. However, to the best of our knowledge, there are no detailed experimental or animal studies in the current literature that clearly explain this relationship. We have added a corresponding statement to the Discussion section to acknowledge this limitation and highlight the need for further basic and translational research to clarify the underlying mechanisms. (page: ,line)

  1. Comments: There are multiple spelling and grammar errors in the writing (such as "toksisity" "rite rion"), which need to be proofread by a native speaker.

Response 7: We thank the reviewer for this helpful comment. We acknowledge the presence of several minor spelling and grammatical errors in the initial version of the manuscript. The manuscript has been scheduled for rapid English editing by the journal’s professional language editing service to ensure full linguistic accuracy before publication.

  1. Comments: The reference list format needs further optimization.

Response 8: We thank the reviewer for this observation. All references have been carefully checked and revised according to the journal’s formatting requirements.Inconsistent citations and numbering errors, if any, have been corrected to ensure accuracy and compliance with the journal style.

We sincerely thank the reviewer for the positive and encouraging comments. We truly appreciate the time and effort dedicated to reviewing our manuscript. All suggested revisions have been carefully addressed, and we believe that these improvements have strengthened the overall quality and contribution of our study.

Round 2

Reviewer 2 Report

Comments and Suggestions for Authors

The author has addressed my concerns. I think the article can be published.